# Purkinje Cell Patterning—Insights from Single-Cell Sequencing

**DOI:** 10.3390/cells11182918

**Published:** 2022-09-18

**Authors:** Elizabeth J. Apsley, Esther B. E. Becker

**Affiliations:** 1Nuffield Department of Clinical Neurosciences, University of Oxford, Oxford OX3 9DU, UK; 2Kavli Institute for Nanoscience Discovery, University of Oxford, Oxford OX1 3QU, UK

**Keywords:** cerebellum, development, Purkinje cell, Zebrin, stripes, transcriptomics, RNA sequencing

## Abstract

Despite their homogeneous appearance, Purkinje cells are remarkably diverse with respect to their molecular phenotypes, physiological properties, afferent and efferent connectivity, as well as their vulnerability to insults. Heterogeneity in Purkinje cells arises early in development, with molecularly distinct embryonic cell clusters present soon after Purkinje cell specification. Traditional methods have characterized cerebellar development and cell types, including Purkinje cell subtypes, based on knowledge of selected markers. However, recent single-cell RNA sequencing studies provide vastly increased resolution of the whole cerebellar transcriptome. Here we draw together the results of multiple single-cell transcriptomic studies in developing and adult cerebellum in both mouse and human. We describe how this detailed transcriptomic data has increased our understanding of the intricate development and function of Purkinje cells and provides first clues into features specific to human cerebellar development.

## 1. Introduction

The cerebellum has a long-established role in motor learning and coordination but is also increasingly implicated in higher cognitive and affective processes [1,2,3]. Its protracted development makes the cerebellum particularly vulnerable to genetic and physical insults, resulting in many different disorders including structural malformations such as Joubert syndrome, movement disorders such as ataxia, neuropsychiatric disorders including autism spectrum disorder, and childhood brain tumors such as medulloblastoma [4,5].

Unlike the cerebral cortex, the cytoarchitecture of the cerebellar cortex is remarkably uniform and arranged in three distinct layers, from outer to inner layer: molecular, Purkinje cell, and granule cell layer. Purkinje cells are central to cerebellar processing as they provide the sole output of the cerebellar cortex. Furthermore, Purkinje cells are the first cerebellar cortical neurons to be born and are considered master regulators of cerebellar development based on their cellular and molecular signaling [6]. While Purkinje cells may at first glance appear to be a uniform cell population in the cerebellum, they exhibit notable differences in their patterns of gene expression. The most well-known of these are the alternating stripes of aldolase C expression, visualized by Zebrin II positivity, in Purkinje cells of the adult cerebellum [7]. Specific gene expression patterns translate into distinct physiological properties of Purkinje cell subtypes and are key to the modular organization of the cerebellum [8].

The vast heterogeneity of Purkinje cells is established early during development. Elucidating the developmental processes that give rise to the functional diversity of Purkinje cells is thus key to our understanding of how the cerebellum performs its numerous functions and may provide important insights into the mechanisms underlying the many diseases associated with aberrant cerebellar development.

For decades, cerebellar development has been studied with imaging and anatomical studies in wildtype and mutant animal models. This work has led to a detailed understanding of the key cell types and their origins (reviewed in [9,10,11]). The recent application of single-cell RNA-sequencing (scRNA-seq) to understand cerebellar development has provided extensive information about the gene expression of distinct cell types in the cerebellum. In addition, transcriptomic analysis of human cerebellar samples is beginning to shed light on human-specific features, which have previously been inaccessible. Here, we will give a brief overview of cerebellar development and then discuss the complexity within the Purkinje cell population from embryonic development to adult. We aim to draw together previous histological studies of marker gene expression with recent scRNA-seq experiments to give an updated description of Purkinje cell heterogeneity in both mouse and human cerebellum.

## 2. Development of the Cerebellum

The cerebellum is one of the first brain structures to develop, but one of the last to achieve its mature configuration [12]. During development, the cerebellum undergoes a dramatic increase in size of more than 1000-fold, which is largely driven by the secondary expansion of granule cell progenitors later during development. During this time, the surface area of the cerebellum increases much more than its volume due to the formation of intricately folded lobules. The resulting brain structure has the highest cell density of any brain area, approximately four times that of the neocortex [13], and despite its misleading name, the “little brain” contains more neurons than the rest of the brain combined. Much has been learned about the development of the cerebellum from research in animals, particularly mice, where cerebellar development takes place over a period of 30–35 days, starting at embryonic day (E) 8.5 with the specification of the cerebellar territory. The main steps in the formation of the cerebellum in mice with a focus on the generation of Purkinje cells are summarized below, before reviewing the contributions of recent single-cell transcriptomic studies to our understanding of cerebellar development in mice and humans.

### 2.1. Early Cerebellar Development

The cerebellum forms from rhombomere 1 adjacent to the midbrain–hindbrain boundary, specified by temporally and spatially coordinated expression patterns of distinct transcription factors. Early transplantation experiments identified the isthmic organizer as a key orchestrator of cerebellar induction via secretion of FGF8 [14,15,16]. The territorial specification of the cerebellar anlage is followed by the generation of cerebellar progenitor cells. All cerebellar neurons develop in overlapping waves from two germinal zones that are specified by the expression of basic helix-loop-helix transcription factors: the rhombic lip (RL) is defined by expression of ATOH1, and the ventricular zone (VZ) is specified by expression of PTF1A (Figure 1A). From E10.5 onwards, ATOH1-expressing cells of the RL produce all glutamatergic neurons, starting with large deep cerebellar nuclei neurons, followed by granule cells and finally unipolar brush cells [17]. In contrast, the PTF1A+ VZ gives rise to all GABAergic neurons: first deep cerebellar nuclei, then Purkinje cells followed by GABAergic interneurons [18,19].

### 2.2. Generation of Purkinje Cells

The PTF1A+ VZ is subdivided into two zones, which give rise to different cell populations (Figure 1B): the OLIG2-expressing posterior region produces SKOR2+ Purkinje cells, whereas the GSX1+ anterior region gives rise to PAX2+ GABAergic interneurons [20]. During development, the OLIG2+ region decreases in size and GSX1+ expression expands until it covers almost the entire VZ [20]. Consequently, in mice Purkinje cells are produced from E10.5 to E13.5 [21], whilst the majority of PAX2+ interneurons are produced later. Post-mitotic Purkinje cell precursors migrate outwards from the VZ along the processes of radial glia to form a layer called the Purkinje cell plate (PCP). Initially around ten cells thick, the PCP forms from E14.5 in mice (Figure 1C). Purkinje cell migration occurs in successive waves with earlier born cells forming the initial plate and being located more dorsally than later born cells [11]. Later, Purkinje cells spread tangentially to form a single-cell layer in the early postnatal cerebellum (Figure 1D). Mouse Purkinje cell development continues for around three weeks after birth. The formation and pruning of afferent connections with climbing fibers and parallel fibers occurs simultaneously with the maturation of Purkinje cells and the formation of their extensive dendritic arbors. The timeline of the physiological and dendritic maturation of Purkinje cells is specific to cerebellar regions, with Purkinje cells in the anterior cerebellum developing more slowly [22].

Human cerebellar development occurs over a much longer time period than in mouse, starting approximately 30 days post conception (dpc) and continuing for up to 2–3 years after birth [23]. The overall developmental process is similar in both species, particularly during the early stages, with mouse embryonic days from E10.5 to E17.5 approximately corresponding to 30–56 dpc (Carnegie stage CS12-23) in humans based on histological observations [24]. However, there are several significant developmental differences between the two species. In humans, the VZ continues to expand for longer and is split into a VZ and a subventricular zone (SVZ), reminiscent of the subdivision of the VZ in the developing cerebral cortex [24]. Similarly, the human RL is both spatiotemporally expanded and compartmentalized [24]. These developmental differences might contribute to the overall increased number of neurons in the human cerebellum and its increased surface area and folial complexity compared to mouse [13]. Recent findings regarding differences in the development of the human cerebellum underscore the importance of gaining better molecular and mechanistic insights into the underlying species-specific features.

### 2.3. Recent Transcriptomic Insights into Purkinje Cell Development

Our understanding of cerebellar development has largely been drawn from anatomical and lineage tracing studies in wildtype and mutant mice. Recently, single-cell transcriptomic analyses of developing and adult samples have provided more detailed profiles of gene expression in both mouse and human cerebellar development (Table 1).

Two recent single-cell RNA-sequencing (scRNA-seq) studies capture the developing cerebellum in mouse and human, respectively [25,26]. Aldinger et al. performed scRNA-seq on human developing cerebellar samples from 9 post conception weeks (pcw) to 21 pcw, identifying 21 transcriptionally distinct cell types [26]. A similar study in mouse by Carter et al. sequenced cerebellar samples from E10 to P10 and identified 15 cell types [25]. Both studies captured the range of cell types expected within the developing cerebellum, including both glutamatergic and GABAergic neurons, such as Purkinje cells, as well as glial cells. Moreover, this work identified differentially expressed genes, which can be useful in distinguishing cell populations and understanding their development. Comparing the transcriptomic profiles of developing mouse and human Purkinje cells confirmed many of the cell-type marker genes that have been previously described, e.g., *RORA*, *FOXP2*, and *CA8* are identified as Purkinje cell markers in both species (Table 2, Figure 2A,B) [25,26]. In addition, less well-characterized genes were found to provide alternative markers for Purkinje cells across both species including *CNTNAP4*, *EBF3*, *BCL11A*, and *EPHA5*. It should be noted that these genes are selected for specificity to Purkinje cells within the cerebellum, but they may also be expressed in other cell types across different brain regions.

**Table 1 cells-11-02918-t001:** Recently published single-cell and single-nucleus RNA-sequencing datasets from the developing and adult cerebellum. E, embryonic days post-conception (mouse, opossum); INTACT, isolation of nuclei tagged in specific cell types; P, days after birth (mouse); pcw, post conception weeks (human); snATAC-Seq, single-nucleus assay for transposase-accessible chromatin sequencing; scRNA-seq, single-cell RNA-sequencing; snRNA-seq, single-nucleus RNA-sequencing.

Reference	Species	Time Points	Sequencing Method (Library Preparation, Sequencing)	Cell Number ^1^	Cell Clusters/Types
Carter et al., 2018 [25]	Mouse	E10–P10	scRNA-seq (10x Genomics, Illumina HiSeq 2500)	39,245	48 clusters, 15 cell types
Peng et al., 2019 [27]	Mouse	P0, P8	scRNA-seq (10x Genomics Chromium, Illumina HiSeqX)	21,119	8 cell types
Rodriques et al., 2019 [28]	Mouse	Adult	Slide-seq (Illumina NovaSeq)	Not specified	12 cell types, focus on comparison of Z+ and Z− stripes
Vladoiu et al., 2019 [29]	Mouse	E10–P14	scRNA-seq (10x Genomics Chromium, Illumina 2500)	62,040	30 cell types
Wizeman et al., 2019 [30]	Mouse	E13.5	scRNA-seq (10x Genomics Chromium, Illumina NextSeq 500)	9326	19 clusters
Aldinger et al., 2021 [26]	Human	9–21 pcw	snRNA-seq ^2^ (SPLiT-seq, Illumina NovaSeq)	69,174	21 cell types
Kozareva et al., 2021 [31]	Mouse	Adult (16 different regions)	snRNA-seq (10x Genomics Chromium)	611,034	46 clusters, 18 cell types
Sarropoulos et al., 2021 [32]	Mouse	E10.5–P63	snATAC-seq (10x Genomics, Illumina NextSeq 550)	91,922	12 broad cell types, 42 subtypes and cell states
	Opossum	P21 & Adult			
Sepp et al., 2021 [33]	Mouse	E10.5–P63	snRNA-seq (10x Genomics Chromium, Illumina HiSeq 4000)	395,736 (115,282 mouse, 180,956 human, 99,498 opossum)	25 cell types, 44 cell states, (12 cell states further split into 49 subtypes)
	Human	7 pcw–adult			
	Opossum	E14–adult			
Chen et al., 2022 [34]	Mouse	Adult	INTACT snRNA-seq (10x Genomics, Illumina NovaSeq6000)	52,487 ^3^	5 broad cell types (focus on 2 subtypes of Purkinje cells: Z+ and Z−)
Khouri-Farah et al., 2022 [35]	Mouse	E12.5–14.5	scRNA-seq (10x Genomics Chromium) snATAC-seq (10x Genomics)	31,144	26 cell types

^1^ Cell number given after quality control. ^2^ Combined with datasets previously generated [36]. ^3^ Additional sequencing was also carried out after behavioral treatments, cell numbers not specified.

Many genes identified as Purkinje cell markers in only one study showed high expression in Purkinje cells across species but did not reach the significance threshold applied for differentially enriched genes. However, some species-specific differences can be revealed by the transcriptomic data. For example, *MDGA1* is identified as enriched in Purkinje cells in the human data [26,33], whilst its expression is low in mouse Purkinje cells [25]. Instead, *Mdga1* is most highly expressed in the mouse granule cell cluster [25]. The latter is consistent with Allen Brain Atlas in situ hybridization data in the developing mouse brain and an alternative developing mouse cerebellar transcriptomic dataset [33]. Together, this indicates a possible species-specific difference in Purkinje cell *MDGA1* expression that warrants further investigation.

Clustering of scRNA-seq data based on transcriptional similarity does not only allow identification of common gene expression patterns within cell types but can also give clues to lineage trajectories. Khouri-Farah et al. reanalyzed published data from mouse embryonic cerebellum [25], supplemented by their own scRNA-seq data, and suggested four developmental trajectories from cerebellar VZ progenitors between E10.5 to E13.5 [35]. These trajectories are distinguished by mutually exclusive expression of genes *Atoh1* and *Ascl1*. In agreement with the established view of cerebellar development, the *Atoh1* branch located at the RL gave rise to glutamatergic neurons and one of the three *Ascl1* trajectories led to the generation of GABAergic neurons. *Ptf1a* was identified as a driver gene in the cascade required for Purkinje cell specification, matching previous experimental studies showing that this transcription factor is essential for Purkinje cell production [18,19]. Whilst Khouri-Farah et al. focus on early stages of Purkinje cell specification, further trajectory analysis of this kind may help to identify driver genes for later stages of Purkinje cell differentiation and cluster specification, which are less well understood.

Single-cell sequencing across species can provide an evolutionary perspective on developmental neurogenesis within the cerebellum. Recent single-nucleus RNA-sequencing (snRNA-seq) was carried out across mouse, human, and opossum cerebellar development and spanning a large range of developmental stages from embryonic to adult [33], thereby complementing earlier studies with developmental [25,26] or adult [31] focus. The availability of multiple cerebellar datasets at single-cell resolution has confirmed findings from individual studies and allows for the identification of more comprehensive and robust transcriptional signatures of developing Purkinje cells. For example, commonly identified Purkinje cell markers from the mouse and human datasets (Table 2) also show high expression in the Purkinje cell cluster identified in developing human and mouse cerebellar data from Sepp et al. [33].

Furthermore, the integration of transcriptomic data across species allows for the calculation of pseudoages as an estimation of equivalent developmental stages. Based on this approach, the cerebellum of a newborn human was inferred to correspond to a one-week-old mouse and a three-week old opossum [33]. This highlights a key difference in the trajectory of human cerebellar development with the majority of cell proliferation occurring in utero in contrast to the postnatal period in the other species [23].

**Table 2 cells-11-02918-t002:** Common Purkinje cell marker genes across mouse and human identified from single-cell sequencing. Differentially upregulated genes in the Purkinje cell cluster were identified in human [26] and mouse [25] transcriptomic datasets using the Seurat FindMarkers function. Human homologues of the mouse Purkinje cell gene set were found using Ensembl BioMart (Ensembl release 107—July 2022 [37]). A total of 73 genes were found in common in Purkinje cells across the two datasets. These are listed in order of average log2(fold change) in the human Purkinje cell cluster. A total of 215 genes were only found as Purkinje cell markers in the human dataset, and 486 only in the mouse dataset. However, many of these showed high expression in Purkinje cells across both species, though not reaching significance. avg_log2FC, Average log2(fold change) in expression in the Purkinje cell cluster compared to all other cells in the dataset; pct.1, Proportion of Purkinje cells expressing gene; pct.2, Proportion of cells expressing gene in the second-highest cluster.

	Gene	avg_log2FC	pct.1	pct.2		Gene	avg_log2FC	pct.1	pct.2
1	*RORA*	2.31025	0.999	0.826	38	*CADM2*	0.500989	0.917	0.766
2	*DAB1*	2.254189	0.993	0.674	39	*NRXN3*	0.493309	0.995	0.758
3	*FOXP2*	1.894787	0.966	0.418	40	*NCAM1*	0.482498	0.949	0.785
4	*CA8*	1.884526	0.895	0.148	41	*SOX4*	0.449056	0.448	0.195
5	*CNTNAP4*	1.685789	0.801	0.171	42	*NPTN*	0.443094	0.463	0.167
6	*AUTS2*	1.605596	1	0.967	43	*DCX*	0.429353	0.556	0.288
7	*EBF1*	1.571646	0.946	0.511	44	*KCTD1*	0.425951	0.466	0.2
8	*EBF3*	1.171463	0.825	0.252	45	*EBF2*	0.409443	0.233	0.084
9	*BCL11A*	1.045316	0.744	0.202	46	*SRGAP1*	0.392077	0.72	0.453
10	*EPHA5*	0.91761	0.727	0.268	47	*DLGAP4*	0.391699	0.533	0.269
11	*NTM*	0.909009	0.985	0.797	48	*FSTL5*	0.379821	0.667	0.373
12	*CELF2*	0.863531	0.987	0.836	49	*PHACTR1*	0.379253	0.796	0.572
13	*DNER*	0.80226	0.773	0.343	50	*UBE2E2*	0.351005	0.686	0.448
14	*MEF2C*	0.769929	0.555	0.099	51	*PKIA*	0.338708	0.48	0.251
15	*LRRC3B*	0.769179	0.566	0.132	52	*PRKAR2B*	0.329412	0.411	0.194
16	*FOXP1*	0.76819	0.65	0.309	53	*SLIT2*	0.327772	0.543	0.317
17	*TENM2*	0.767349	0.95	0.755	54	*KIDINS220*	0.324566	0.61	0.389
18	*PPP2R2B*	0.764795	0.83	0.447	55	*ZNRF1*	0.314214	0.431	0.209
19	*KCNIP1*	0.759729	0.744	0.306	56	*CADM1*	0.313522	0.624	0.413
20	*ATP2B2*	0.736813	0.65	0.247	57	*SYT16*	0.310313	0.268	0.058
21	*EPHA7*	0.736577	0.685	0.365	58	*MYT1L*	0.306548	0.901	0.65
22	*FOXP4*	0.707523	0.461	0.046	59	*ARHGAP20*	0.291987	0.269	0.093
23	*XPR1*	0.700962	0.724	0.343	60	*WNT7B*	0.29075	0.216	0.026
24	*PCDH17*	0.690897	0.566	0.187	61	*CACNG2*	0.290066	0.339	0.14
24	*CNTN5*	0.667829	0.928	0.687	62	*LYPD1*	0.289998	0.221	0.03
26	*NRP2*	0.667035	0.434	0.077	63	*GNG2*	0.289336	0.481	0.276
27	*KITLG*	0.659408	0.455	0.091	64	*CRMP1*	0.282869	0.489	0.28
28	*EPHA4*	0.653552	0.482	0.124	65	*GAD1*	0.275746	0.351	0.128
29	*RBMS1*	0.627325	0.595	0.207	66	*CMIP*	0.267756	0.653	0.46
30	*CTNNA2*	0.6253	0.982	0.801	67	*PCP4*	0.264464	0.206	0.04
31	*ANK2*	0.60591	0.966	0.724	68	*NSG1*	0.263876	0.257	0.076
32	*CTTNBP2*	0.58162	0.609	0.232	69	*SLC1A6*	0.261146	0.197	0.029
33	*SPOCK3*	0.56822	0.538	0.215	70	*RAB3C*	0.260151	0.373	0.215
34	*NTRK3*	0.531684	0.772	0.481	71	*CHD3*	0.25663	0.349	0.165
35	*PID1*	0.516531	0.44	0.174	72	*NNAT*	0.256533	0.346	0.182
36	*LHX1*	0.512295	0.402	0.073	73	*EVL*	0.251227	0.529	0.335
37	*MACROD2*	0.511606	0.889	0.667					

In addition to gene expression, single-cell transcriptomic studies give an estimate of the proportion of different cell types across cerebellar development. As expected, progenitors and Purkinje cells were found to be highest in early development, before the clonal expansion of granule cell progenitors, which dominate in later periods [33] (Figure 2C). Interestingly, multiple transcriptomic studies found an approximately two-fold increase in the peak proportion of Purkinje cells in human cerebellum compared to mouse [25,26,33] (Figure 2D). This increase in the proportion of Purkinje cells is likely to correspond to the presence of a SVZ containing mitotic progenitors in the human VZ, in contrast to mice where mitosis and terminal differentiation only occur at the most ventricular edge [24]. The human SVZ showed significant expansion of progenitors between CS18 to CS23 (around 44–56 dpc) accompanying high levels of differentiation of VZ progenitors [24]. This period approximately matches the peak proportions of Purkinje cells in the human scRNA-seq datasets at 8–9 pcw.

While scRNA-seq captures the pattern of gene expression defining cell identities, tracking changes at chromatin level provides greater understanding of the regulation of cell states. snATAC-seq (single-nucleus assay for transposase-accessible chromatin using sequencing) on embryonic mouse cerebellar samples has identified transcription factors important in defining cell states in the developing cerebellum [32,35]. Regions of chromatin with differentially increased accessibility in Purkinje cells showed an enrichment for binding sites of homeodomain transcription factors including EN1 and LHX4 [32,35]. This approach might help to identify key drivers of specific developmental stages that could be further explored functionally. In addition, the integration of chromatin and transcriptomic data allows the classification of transcription factors as being broadly activators or repressors depending on positive or negative correlation of accessibility peaks at their binding sites to nearby gene expression. Khouri-Farah et al. identified 33 putative activators and 16 repressors, with the latter including members of the FOXP family [35]. It will be interesting to elucidate the transcriptional targets of these to better understand the mechanisms underlying Purkinje cell development.

## 3. Purkinje Cell Complexity

Despite the uniform appearance of the cerebellum with regards to its three-layered architecture and stereotypic wiring, the cerebellum is highly compartmentalized and made up of distinct cell subtype identities and connections, reflecting a vast array of spatially specialized functions. Spatial diversity has long been recognized in the Purkinje cell population, although we are only beginning to understand the underlying molecular identities and physiological significances. In this section, we will briefly summarize the complexity of Purkinje cells as it is currently known in the adult and developing cerebellum and then explore what we can learn from recent transcriptomic studies at the single-cell level.

### 3.1. Adult Cerebellar Topography

The adult cerebellum is folded horizontally to form ten lobules, which can be grouped into four transverse zones: the anterior zone (AZ—lobules I-V), central zone (CZ—lobules VI–VII), posterior zone (PZ—lobules VIII-IX), and nodular zone (NZ—lobule X) (Figure 1E and Figure 3A). Broadly, the anterior lobules of the cerebellum act in sensorimotor functions, whereas the posterior regions connect to regions such as the prefrontal and temporal cortex and thought to be involved in higher cognitive functions [2]. Regional differences are also evident in a range of cerebellar pathologies [8,38]. Greater vulnerability and cell death have been frequently described in the anterior cerebellum, for example in mouse models and patients with autosomal-recessive spastic ataxia of Charlevoix-Saguenay (ARSACS) [39] as well as in Niemann–Pick type C disease [40]. In contrast, some mutants such as the nervous mouse show selective cell death of posterior regions [8,38]. It should be noted that parasagittal patterning differs across zones and lobules (Figure 3A). Therefore, regionalized pathologies likely reflect greater vulnerability of cells belonging to a particular molecular pattern rather than a transverse zone per se.

In addition to the transverse zones, the adult cerebellum is patterned by parasagittal stripes that are highly conserved through evolution [43] (Figure 3A). Stripes are defined by the pattern of expression of Aldolase C (*Aldoc*), also known as Zebrin II, in Purkinje cells [7]. In the mouse cerebellum, there are seven Zebrin-positive (Z+) stripes, which are arranged symmetrically to either side of the midline (Zebrin stripes have been described in detail in [41]). Other markers such as the small heat shock protein HSP25 also show parasagittal striped expression in Purkinje cells in the CZ and NZ areas, where all Purkinje cells are Z+, revealing further complexity [44]. Many markers are co-localized with either Z+ or Z− Purkinje cells. For example, PKCδ, EAAT4, PLCβ3, NCS1, GABABR2 are expressed in the Z+ population, whereas others including PLCβ4, mGluR1b, and NRGN are expressed in a complementary pattern in the Z− cells [8]. Whilst Zebrin is most commonly used to define parasagittal stripes, studies have also used NFH, the antibody P-path and Parvalbumin to describe Purkinje cell stripes [45,46,47]. Overall, the cerebellar cortex is composed of more than 200 parasagittal stripes [48].

At a higher resolution, the cerebellum is functionally subdivided into modules, which form the basic operational processing unit of the cerebellum. A cerebellar module is defined as a longitudinal, i.e., (para)sagittally organized, zone of Purkinje cells in the cerebellar cortex that receives common climbing fiber input from a particular region of the inferior olive. In turn, the same Purkinje cells target a discrete part of the cerebellar nuclei, thereby creating a closed loop [41]. Modules typically contain only a few hundred Purkinje cells [48]. Different modules display different physiological properties, which may enable different forms of learning and activity. Comparison of parasagittal stripes has found that Z− Purkinje cells have increased intrinsic excitability and a higher rate of both simple and complex spike firing than Z+ cells [49,50]. These electrical properties are cell-intrinsic and in Z− cells at least partially mediated by TRPC3 activity [50,51].

Given the implications of Purkinje cell topology for cerebellar structure, function, and pathology, it is important to understand the origins of this intricate patterning.

### 3.2. From Cells to Clusters to Stripes

Purkinje cell subtype specification occurs soon after Purkinje cells are born through a complex reorganization into first clusters and then stripes. Birth dating studies in mice have identified that Purkinje cells generated at E10.5, E11.5, and E12.5, respectively, give rise to specific medio-lateral clusters [21] (Figure 3B). The mechanisms through which distinct subgroups of Purkinje cells are specified have been the subject of intense scrutiny.

After migration from the VZ, Purkinje cells form the thick PCP, and nine different cell clusters can be distinguished based on the expression of molecular markers including *Pcdh10* and *Epha4*, and variable levels of *Foxp2* [52,53] (Figure 3C,D). In general, the more lateral and dorsal clusters contain a higher proportion of earlier-born Purkinje cells (E10.5, E11.5) whilst the more medial and ventral clusters contain more later-born Purkinje cells (E11.5-E13.5). However, the three-dimensional cell arrangement is more complex than just an age gradient, with most clusters containing a mixture of Purkinje cells born at different times.

Purkinje cell progenitors were thought to migrate directly outwards to the dorsal surface along radial glia fibers [54]. However, detailed study of the initial formation of the PCP at E14.5 indicates some early-born Purkinje cell progenitors migrate tangentially from the posterior VZ before undergoing a Reelin-dependent change in direction to radial migration by E14.5 [55]. This suggests that the migration of Purkinje cell progenitors can differ depending on spatial and timing factors, thereby enabling the formation of separate aggregates or clusters. From E14.5, embryonic clusters transform and increase in number [53]. Different clusters are separated by narrow gaps called granule cell raphes, which contain no Purkinje cells and are later filled by migrating granule cells [42]. Over 50 embryonic clusters have been identified at E17.5 based on a combination of marker genes: *Foxp2* labels most Purkinje cells [56], whereas the expression of other markers including *Plcb4*, *Epha4*, *Pcdh10*, and *Itpr1* distinguish spatially distinct Purkinje cell clusters [42,53] (Figure 3D). Determining the molecular identity of Purkinje cell clusters using imaging is limited by the number of employed probes. For example, other genes that were later identified to be differentially expressed across the developing cerebellum such as *Foxp1* [30] were not used in these studies (Figure 3D).

Between around E18 and P20 embryonic clusters transition to adult stripes, mediated by Reelin secretion by the external granule layer (EGL) [57,58]. This is accompanied by thinning out of the PCP to form a single-cell layer. Embryonic clusters can be tracked to the formation of distinct stripes at P6 [42], with each cluster contributing to one or multiple adult stripes [59]. As the cerebellum grows predominantly in the sagittal plane, forming the folded lobule structure of the adult cerebellar cortex, the clusters become stretched to long parasagittal stripes. Many of the marker genes that show differential expression in embryonic Purkinje cell clusters are expressed broadly in adult Purkinje cells or are localized in subpopulations that no longer express the same marker combination as earlier in development. For example, *Pcp2* (L7) is only expressed in a pair of medially located clusters to either side of the midline at E14.5 and then in three pairs of clusters at E15.5 but is subsequently expressed broadly in adult Purkinje cells [59]. Mapping of the relationships between embryonic clusters to the pattern of adult stripes is challenging, due to a gap in time between expression of many embryonic and adult markers. However, some markers such as *Plcb4* are expressed from embryonic development through to the adult cerebellum and can be used to track clusters as they undergo transformations. Engrailed homeobox genes *En1* and *En2* are important in defining mediolateral patterning in the developing cerebellum [60,61]. These transcription factors act in the differential formation of lobules in the vermis, the central structure of the cerebellum, and the surrounding hemispheres [60]. In addition, *En1/2* genes influence the formation of Zebrin and HSP25 parasagittal stripes and themselves show a related spatial distribution of expression [61,62].

### 3.3. Understanding Purkinje Cell Identity Using Single-Cell Transcriptomics

Recent single-cell transcriptomic datasets of developing cerebellar samples have given further insight to embryonic Purkinje cell subtypes in both mouse and human. Five spatially distinct Purkinje cell subtypes were identified in the E13.5 mouse cerebellum, marked by high expression of *Etv1*, *Nrgn*, *En1*, *Cck*, and *Foxp1*, respectively [30]. All of the identified Purkinje cell subtypes broadly expressed *Foxp2* [30], but with differing levels of expression matching previous observations [52,56]. Interestingly, a second developing mouse study identified similar embryonic Purkinje cell subtypes with overlap in key genes, despite integrating snRNA-seq samples from a wider developmental time window [33]. Four subtypes were identified based on differentially upregulated expression of marker genes *Rorb*, *Cdh9*, *Foxp1*, and *Etv1*, respectively [33] (Figure 4A). These subtype marker genes label spatially distinct Purkinje cell clusters (Figure 4B). Many of the variably expressed markers used for initial histological studies of embryonic Purkinje cell clusters including *Itpr1*, *Pdch10*, and *Epha4* show distinct and differential patterns of expression across the four Purkinje cell subtypes (Figure 4A). Interestingly, the subtypes correlated spatially with *Ebf1* expression (high in medial vs. lateral) and temporally with *Ebf2* expression (high in late vs. early) [33] (Figure 4A). Thus, both studies identified an early, laterally located subtype of Purkinje cells with high *Foxp1* and low *Ebf2* expression. Other clusters also appear to correlate: a subtype labelled by high *Etv1* and medium *Ebf2* levels is present in both datasets; both also share a medially located *Calb1/Cdh9* subtype with high *Cdh9* expression (Figure 4C). Together, multiple RNA-seq studies suggest the presence of spatially distinct Purkinje cell subtypes in embryonic mice, with differing expression of genes including *Ebf1*, *Ebf2*, *Foxp1*, and *Etv1*.

The Carter et al. scRNA-seq study of the developing mouse cerebellum did not distinguish different Purkinje cell subtypes [25]. In order to explore the heterogeneity within the Purkinje cells in this data, we performed separate dimension reduction analysis on the Purkinje cell cluster. Using the variable genes, cells were plotted based on the differences in gene expression (Figure 4D). Purkinje cells were separated by age with the postnatal cells forming a cluster expressing more mature markers including *Itpr1* (Figure 4D). Populations of embryonic Purkinje cells clustered into cells with high *Foxp1* or *Ebf2* expression, respectively (Figure 4D), corroborating the other transcriptomic studies [30,33].

In contrast to the diversity of Purkinje cells identified in the developing mouse cerebellum, scRNA-seq analysis of the developing human cerebellum identified only two subtypes of embryonic Purkinje cells that were distinguished by high or low expression of *EBF1* and *EBF2* [33]. Further studies of developing and adult human cerebellar samples are needed to confirm this difference and evaluate the relevance of mouse Purkinje cell subtypes to our understanding of human cerebellar development and disease. Methods to enrich specific cell types may be required to obtain sufficient Purkinje cells to study the heterogeneity within this cell type, particularly in later samples where granule cells dominate in the cerebellum.

Genes differentially expressed between embryonic Purkinje cell subtypes may give insight into functional differences as well as into how these distinct clusters form. For example, the Cadherin family of adhesion molecules was enriched in the variable genes across embryonic Purkinje cell subtypes [33]; the *Cdh9* subtype showed high expression of *Cdh10*, *Cdh12*, and *Cdh18*, whereas the *Rorb* subtype had high expression of *Cdh4*. This is consistent with earlier studies that identified expression of different subsets of cadherins in distinct Purkinje cell clusters [63]. The adhesion molecule profiles of cells may influence how cells migrate with cells expressing common Cadherin types aggregating together, whilst those with different signatures move independently. Thus, distinct cell–cell interactions may enable clusters to form, and changes may direct the transition from embryonic clusters to adult stripes.

The aforementioned scRNA-seq studies characterizing Purkinje cell subtypes lack adult cells and thus, these cannot be employed to map the transition between early and mature Purkinje cell subtypes. However, a recent study characterizes the adult mouse cerebellum in detail, providing expression profiles of adult Purkinje cell subtypes that can be compared to the embryonic subtypes [31]. A total of 16 regions across adult mouse cerebellum were sampled to examine distribution of cell types and subtype identities throughout the lobules of the vermis and hemispheres of the cerebellar cortex [31]. A total of 18 cell types were identified with multiple subclusters for some cell types, reiterating the cellular heterogeneity present in the cerebellum [31]. Of the nine identified Purkinje cell clusters, eight had significantly different distributions across different regions indicating spatial specialization [31]. Purkinje cell clusters were also distinguished by Aldolase C expression with seven clusters being Z+ and only two Z−, suggesting that there may be greater diversity amongst Z+ Purkinje cells [31].

Sepp et al. used correlation of differentially expressed genes to map their embryonic Purkinje cell clusters to the adult Purkinje clusters [31,33] (Figure 4A). Both early-born subtypes correlate most strongly to Z+, more laterally located clusters in the adult cerebellum. In contrast, later-born subtypes were predicted to give rise to Z+ and Z− cells: the Cdh9 subtype correlates with Z+ cells located in medial posterior regions and the Etv1 subtype with Z− cells. These conclusions are somewhat consistent with previous descriptions implicating birth date and *Ebf2* expression in determination of Zebrin II type; Z+ cells are thought to derive from early-born EBF2- cells, whilst Z− cells stem from later born EBF2+ cells [64]. EBF2 has been shown to act by suppressing the Z+ phenotype without driving the Z− phenotype, as ectopic Z+ cells in *Ebf2* null mice show expression of markers associated with both Z+ and Z− phenotypes [65,66]. However, the *Cdh9*-expressing embryonic Purkinje cell subtype suggests that cells with high *Ebf2* expression can also give rise to Z+ cells [33] (Figure 4A). This is consistent with lineage tracing that demonstrated that all Purkinje cells derive from progenitors that have previously expressed *Ebf2* [67]. This may indicate a loss of *Ebf2* expression in the early born Purkinje cells, although the underlying mechanism remains to be explored. While *Ebf2* appears to be a significant gene in Purkinje cell specification, there are likely other genes involved in driving the Z+ phenotype or preventing its suppression by *Ebf2* expression.

While scRNA-seq provides rich information on the gene expression of individual cells, the physical location of each cell in the tissue, and hence the relationship between cell microenvironment and gene expression, is lost using this method. Gene signatures from scRNA-seq analysis can be retrospectively correlated to spatial locations based on imaging of selected markers [30,33]. Alternatively, new methods of spatial transcriptomics address this problem by encoding spatial information about cells prior to scRNA-seq by using barcoded DNA primers fixed onto a glass slide [68] or beaded surface [28]. Applying Slide-seq to adult mouse cerebellum identified spatially defined subpopulations of different cell types [28]. Detection of genes enriched in the Purkinje cell layer identified 126 genes, which showed strong correlation or anti-correlation with the Zebrin pattern, suggesting candidates for further investigation into the functional differences between Z+ and Z− Purkinje cells. In addition, lobules IX and X had more distinct expression patterns, which may reflect the more diverse functions of the posterior cerebellum [28].

Recent work has begun to exploit single-cell transcriptomics to understand activity-induced changes in gene expression in Purkinje cell subpopulations during motor behavior and learning [34]. Isolation of nuclei tagged in specific cell types (INTACT) by flow cytometry was used to enrich for genetically labelled Purkinje cells, followed by snRNA-seq. The isolated Purkinje cells clustered into Z+ and Z− populations, reflecting the well-characterized Zebrin parasagittal stripes of the adult cerebellum. Interestingly, only the Z− Purkinje cells displayed robust gene expression changes after motor activity or leaning and were shown to have a crucial role in associative leaning mediated by FGF2R signaling [34]. Similar approaches in the future are destined to identify functional roles of other Purkinje cell populations such as those involved in specific cognitive and affective processes.

## 4. Conclusions and Perspectives

Recent single-cell transcriptomic studies have provided greater insight into the molecular heterogeneity present in the different cell populations of the developing and adult cerebellum. With regards to Purkinje cell development, new transcriptomic findings are extending previous lineage tracing and cell marker identification studies to define Purkinje cell subgroups and increase our understanding of how embryonic clusters transform to adult stripes. Further functional studies will be required to unravel the mechanisms of how distinct expression profiles of Purkinje cell subtypes relate to functional differences. To date, many studies investigating the physiological properties of Purkinje cells focus only on the binary division of Z+ and Z− Purkinje cells. Little is known about the functional differences between Purkinje cell subtypes at the increased level of complexity that has been uncovered through scRNA-seq studies. Moreover, where functional differences have been identified, for example firing rate in Z+ and Z− stripes in the adult cerebellum, it is unclear if the markers associated with these stripes have biological relevance to the operation of the respective cerebellar module. The combination of genetically labelling specific subtypes, followed by INTACT RNA-seq in the context of specific behaviors is one powerful technique to identify the functional relevance of Purkinje cell subclusters. In addition, RNA-seq can be directly combined with electrophysiological measurements of individual cells using Patch-seq methods [69]. Patch-seq methods have been applied to brain slices and cultured neurons, but not yet in the context of the cerebellum. Another promising area of method development is spatial transcriptomics which will be particularly useful in studying the spatial distribution of Purkinje cell subtypes in clusters and stripes [28].

Single-cell sequencing studies are beginning to shed light on human cerebellar development [26,33]. The human cerebellum has a much larger pool of early Purkinje cells than mice [33], likely due to the expansion of the area of mitosis in the VZ [24]. However, sequencing studies have found only limited subtype specialization in human Purkinje cells, defining only two classes in embryonic samples [33]. Further investigations across developmental time will be key to understand the molecular and functional diversity of Purkinje cells in the human cerebellum and how this relates to knowledge gained from mouse and other species. Moreover, similar studies in diseased cerebellum are poised to identify pathological mechanisms and help us understand how the cerebellum contributes to neurodevelopmental disorders. One of the major challenges is access to human cerebellar tissue. This might be addressed through the use of protocols to differentiate cerebellar neurons and organoids from human induced pluripotent stem cells (hiPSCs) [70,71,72,73]. Human iPSC-derived cerebellar models hold immense potential to model cerebellar development and disease. However, current methods produce only very immature neurons, and it remains to be investigated whether the complexity of human cerebellar development can be recapitulated in vitro. Further RNA-seq studies of human samples will be essential to inform the development and validation of hiPSC-derived models before these will be established.

## Figures and Tables

**Figure 1 cells-11-02918-f001:**
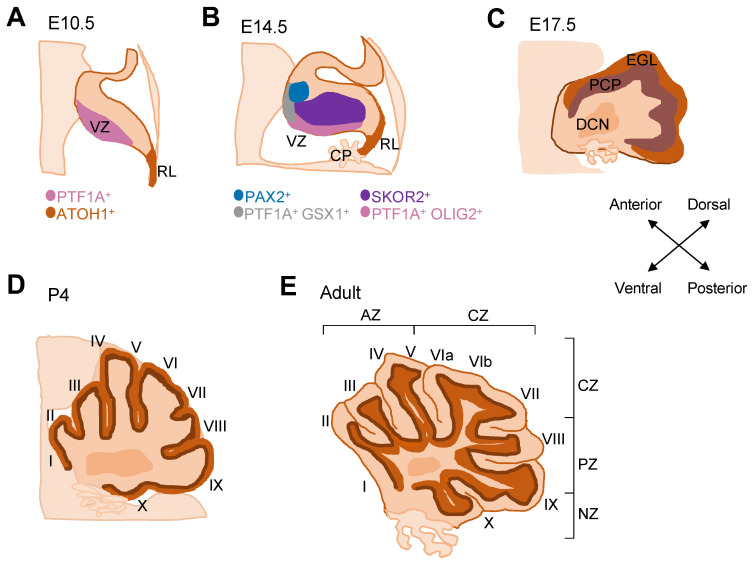
Schematic of mouse cerebellar development. Mid-sagittal sections show the stages of mouse cerebellar development over time from E10.5 to adult. (**A**) At E10.5, two germinal zones are established: the VZ and the RL, which are identified by expression of PTF1A and ATOH1, respectively. (**B**) SKOR2+ post-mitotic Purkinje cells are generated from PTF1A+ OLIG2+ progenitors in the VZ. The VZ also gives rise to PAX2+ interneurons from PTF1A+ GSX1+ progenitors. (**C**) From E14.5, Purkinje cells migrate outwards from the VZ to form the PCP. The PCP is initially several layers of cells thick and lies underneath the appearing EGL. (**D**) In early postnatal stages, the PCP spreads to form the characteristic monolayer of Purkinje cells that exists in the adult. (**E**) The adult cerebellum consists of ten lobules (I–X) from anterior to posterior that are grouped into four transverse zones. The cerebellar cortex has three layers, from the outside: molecular layer, Purkinje cell layer, granule cell layer. AZ, anterior zone; CP, choroid plexus; CZ, central zone; DCN, deep cerebellar nuclei; E, embryonic day; EGL, external granule layer; NZ, nodular zone; P, postnatal day; PCP, Purkinje cell plate; PZ, posterior zone; RL, rhombic lip; VZ, ventricular zone.

**Figure 2 cells-11-02918-f002:**
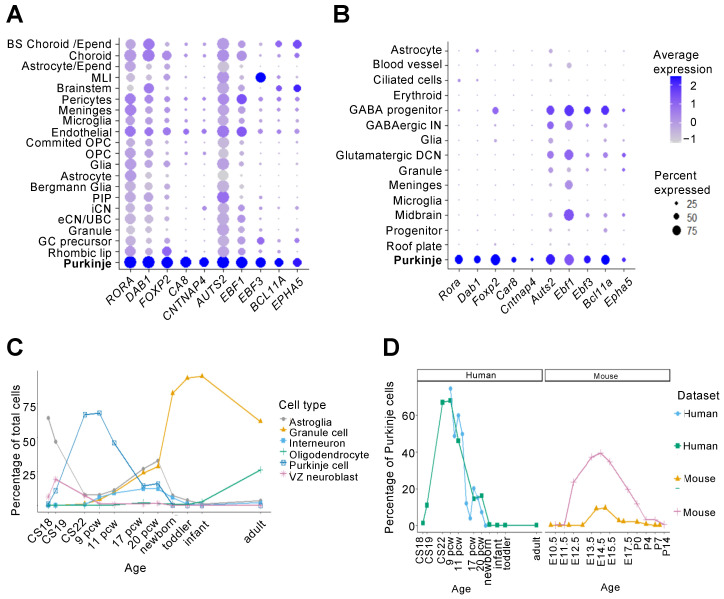
Single-cell transcriptomics of the developing human cerebellum identifies cell type markers and highlights changes in cell populations over time. Expression of Purkinje cell markers is consistent across two transcriptomic datasets. Expression of the top 10 common Purkinje cell markers (Table 2) are shown across cell types in (**A**) Human [26] and (**B**) Mouse [25] developing cerebellar datasets. (**C**) The proportion of different cerebellar cell types changes over the course of human cerebellar development [33]. Only the six cell types with highest frequencies are shown for clarity. (**D**) The developing human cerebellum has an approximately two-fold higher peak percentage of Purkinje cells compared to mouse. This observation is consistent across human (blue circle [26], green square [33]) and mouse (orange triangle [25], pink cross [33]) RNA-seq datasets. BS, brain stem; CS Carnegie stage; DCN, deep cerebellar nuclei; E, embryonic day; eCN, excitatory cerebellar interneurons; Epend, Ependymal; iCN, inhibitory cerebellar nuclei; IN, interneuron; MLI, molecular layer interneurons; OPC, oligodendrocyte progenitor cells; P, postnatal day; pcw, post conception week; PIP, PAX2+ interneuron progenitors; UBC, unipolar brush cell; VZ, ventricular zone.

**Figure 3 cells-11-02918-f003:**
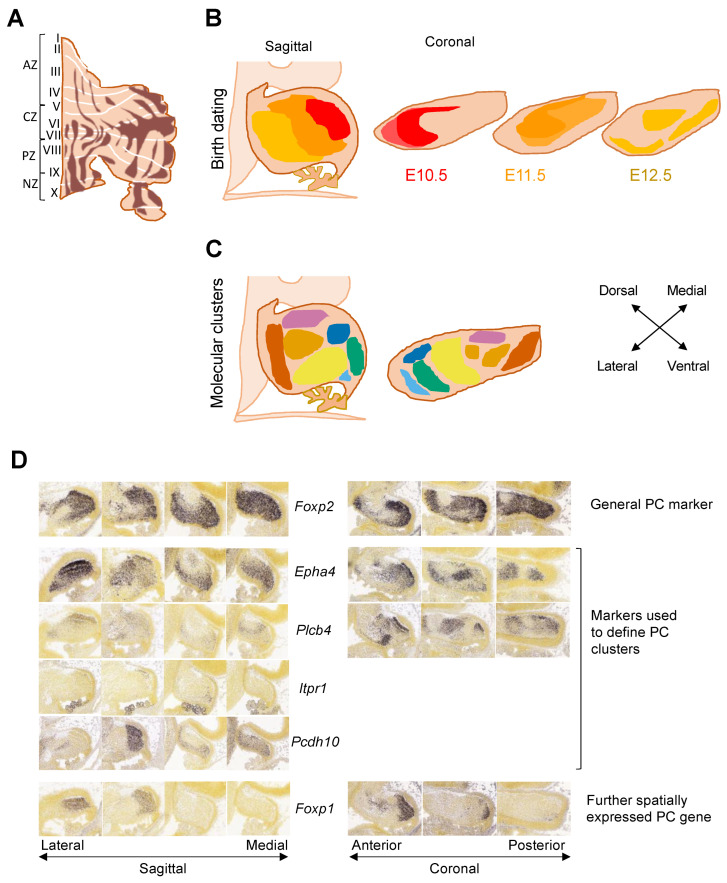
Purkinje cell embryonic clusters defined by imaging and birth dating studies in mice. (**A**) Zebrin stripes define Purkinje cell populations in the adult cerebellum. Stripes are parasagittal with most of the Z+ cells located in the posterior cerebellum. A complete description of the zebrin map is provided in [41] (**B**) E15.5 clusters can be defined by lineage tracing of different Purkinje cell birthdates E10.5 (red), E11.5 (orange), E12.5 (yellow) or (**C**) by expression of different combinations of molecular markers. Colors reflect different molecular clusters identified by [42]. Only one hemisphere of the cerebellum is shown in coronal diagrams. (**D**) Distribution of Purkinje cell markers at E15.5: *Foxp2* is broadly expressed in most Purkinje cells. *Epha4*, *Plcb4*, *Pcdh10*, and *Itpr1* are differentially expressed and often used as molecular markers to define distinct clusters. *Foxp1* is included as an additional marker showing spatially specific expression in the anterior and lateral developing cerebellum. Images of sagittal and coronal sections from Allen Developing Mouse Brain Atlas (https://developingmouse.brain-map.org/ accessed on 7 July 2022) are shown across lateral-medial and anterior-posterior axis. Coronal sections were not available for *Pcdh10* and *Itpr1*. A, anterior; AZ, anterior zone; CZ, central zone; E, embryonic day; L, lateral; M, medial; P, posterior; PC, Purkinje cell; PZ, posterior zone; NZ, nodular zone.

**Figure 4 cells-11-02918-f004:**
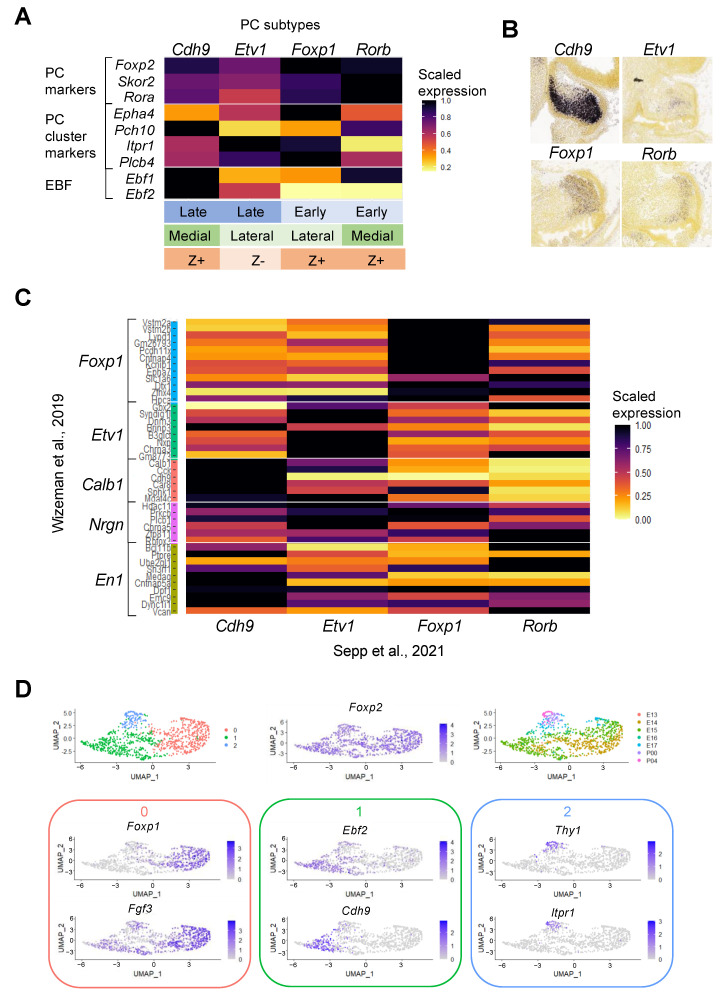
Mouse embryonic Purkinje cell clusters defined by single-cell transcriptomics. (**A**) Sepp et al. define four Purkinje cell subtypes based on transcriptomic differences and labelled by differentially expressed markers *Chd9*, *Etv1*, *Foxp1*, and *Rorb* [33]. All subtypes show broad expression of Purkinje cell markers (*Foxp2*, *Skor2*, *Rora*) but variable expression of genes previously used in imaging studies to define clusters (*Epha4*, *Pcdh10*, *Itpr1*, *Plcb4*) [52,53]. The different subtypes can also be defined by the combination of *Ebf1* and *Ebf2* expression. In addition, subtypes differ with regards to birth date, location, and predicted adult Zebrin phenotype [33]. (**B**) Representative images of key marker gene expression in E15.5 sagittal cerebellar sections corresponding to Purkinje cell clusters in (A). *Cdh9* and *Rorb* are shown in medial sections and *Foxp1* and *Etv1* in lateral sections. ISH images taken from the Allen Developing Mouse Brain Atlas (https://developingmouse.brain-map.org/ accessed on July 2022). (**C**) Correlation between embryonic Purkinje cell subtypes identified by single-cell transcriptomic studies of the developing mouse cerebellum. Wizeman et al. found genes differentially expressed by five Purkinje cell subtypes [30]. The average expression of these genes is shown across the four Purkinje cell subtypes identified by Sepp et al. [33]. Expression of each gene is normalized across samples. (**D**) Dimension reduction analysis of the Purkinje cell cluster from a third developing mouse cerebellar scRNA-seq dataset [25] suggests similar distinct molecular identities to previously reported [30,33]. Top left: A UMAP projection at resolution 0.2 indicates three clusters within the Purkinje cells. Top center: *Foxp2* is expressed across all Purkinje cells. Top right: Across UMAP_2 (y axis) cells are separated by sample date into two embryonic clusters and one postnatal cluster. The embryonic clusters show distinct expression patterns of *Foxp1* and *Ebf2*. The postnatal cells show expression of more mature Purkinje cell markers such as *Thy1* and *Itpr1*. Expression of the top two markers for each cluster are shown.

## Data Availability

Not applicable.

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
