# Peer review of "Purkinje Cell Patterning—Insights from Single-Cell Sequencing"

_cells, 2022, doi:10.3390/cells11182918_

Round 1

Reviewer 1 Report

In this review manuscript, Apsley and Becker provide an outstanding and scholarly account of cerebellar development, patterning, and recent data that have pushed the field and ideas forward with single-cell sequencing approaches. Central to the review is the well-established and long-standing work on cerebellar zones (stripes and modules), although here the authors frame the discussion around the mechanistic issues, primarily molecular, that remain unknown and of high interest in the field. The article is well-written, thorough, and a pleasure to read. I have only minor suggestions below that will help improve the clarity of the main message.

Comments:

1)    In the Abstract, the authors state “…including Purkinje cell subtypes, based on knowledge of a limited number of markers”. Although I get the comparison here in relation to sequencing data, I think it is probably fair to give previous work more credit. In fact, Purkinje cell subtypes have been described with a pretty decent number of markers. My concern is that the current statement undersells the large amount of knowledge gained from previous marker analyses.

2)    The authors state “…This work has led to a good understanding of the key cell types and their origins…” Here again, I think the word choice somewhat downplays the massive amount of knowledge and understanding previously gained in the cerebellar patterning field. I feel that the understanding was better than “good” even before sequencing.

3)    The authors state “However, the recent application of single-cell RNA-sequencing (scRNA-seq) to understand cerebellar development has provided much more detailed information the gene expression patterns in distinct cell types in the cerebellum…” I would argue that the sequencing has given us more information on the actual genes themselves, I am not sure that we have learned much more about patterning per se (at least not yet). This claim needs to be toned down.

4)    In the statement above, the sentence should read “…information about the…”. The word “about” is missing.

5)    The authors state “The cerebellum forms at the midbrain-hindbrain boundary…” I think this is not quite accurate. The cerebellum develops from the hindbrain, which is adjacent to the boundary.

6)    The cardinal lobes of the cerebellum should be labeled in Figure 1D.

7)    The authors state “…The anterior cerebellum frequently shows greater vulnerability and cell death,…” I think this issue requires a more in-depth discussion. It’s not quite so simple because many such models that describe anterior cerebellum defects do so based on sagittal sections. This of course ignores the zonal patterning effects. And because these descriptions are mainly from animal models of ataxia, which tend to have the same patterning issues, it seems like the anterior cerebellum is more affected. But in fact there is a bias in what has been described and how it has been described. The authors correctly allude to the nervous mutant, which has a more pronounced degeneration in zebrin positive cells, as opposed to the ataxia mutants which have degeneration in zebrin negative cells (making the anterior seem more affected, but in its really an issue of zonal patterning rather an issue of anterior versus posterior). Therefore, the authors are certainly not wrong in their statements, but this idea needs to be fully discussed so that those not familiar with the details get a full appreciation of the patterning and pathology.

8)    Given that zones/zebrin patterning is at the heart of this review, the authors would serve the reader well by providing a bigger and more complete schematic of the zebrin map. Figure 3A is very hard to interpret. No stripe specific labels are provided, and the labels for AZ, CZ, PZ, and NZ are hard to match for the hemispheres and vermis since an unfolded map is presented.

9)    Figure 3B needs some labels.

10) For the middle panel of Figure 3D, why not just show both sides of the cerebellum otherwise the coordinates in the right panel are very confusing to interpret.

11) The authors state “the adult cerebellum is patterned by vertical parasagittal stripes”. I am not sure I have ever heard them described as vertical. It might be better to use previous language otherwise readers are likely to get confused.

12) I am not sure I understand what the authors mean by “Reelin-dependent ‘change of posture’.

13) There are several papers from Alexandra Joyner’s lab regarding Engrailed1 and Engrailed2 patterns and how mutations in these genes impact stripe patterning that should be cited, potentially in the second paragraph of page 12.

14) In Figure 4A, what is meant by a “variable” PC marker? I suspect variable isn’t the perfect choice of word.

15) The authors state “To date, many studies investigating the physiological properties of Purkinje cells focus only on the binary division of Z+ and Z- Purkinje cells.”. True, but I think it is also fair to mention the papers that don’t only focus in this. The authors should reference some of the work by Richard Hawkes (p-path, HNK1) and Sillitoe (NFH).

Author Response

Reviewer #1

In this review manuscript, Apsley and Becker provide an outstanding and scholarly account of cerebellar development, patterning, and recent data that have pushed the field and ideas forward with single-cell sequencing approaches. Central to the review is the well-established and long-standing work on cerebellar zones (stripes and modules), although here the authors frame the discussion around the mechanistic issues, primarily molecular, that remain unknown and of high interest in the field. The article is well-written, thorough, and a pleasure to read. I have only minor suggestions below that will help improve the clarity of the main message.

Comments:

1) In the Abstract, the authors state “…including Purkinje cell subtypes, based on knowledge of a limited number of markers”. Although I get the comparison here in relation to sequencing data, I think it is probably fair to give previous work more credit. In fact, Purkinje cell subtypes have been described with a pretty decent number of markers. My concern is that the current statement undersells the large amount of knowledge gained from previous marker analyses.

We have rephrased this sentence in the abstract as “knowledge of selected markers” to avoid giving the impression of underselling previous studies.

2) The authors state “…This work has led to a good understanding of the key cell types and their origins…” Here again, I think the word choice somewhat downplays the massive amount of knowledge and understanding previously gained in the cerebellar patterning field. I feel that the understanding was better than “good” even before sequencing.

We have rephrased this sentence as ‘detailed understanding’ and also edited other statements in the same paragraph (lines 47-58) to address the reviewer’s concerns.

3) The authors state “However, the recent application of single-cell RNA-sequencing (scRNA-seq) to understand cerebellar development has provided much more detailed information the gene expression patterns in distinct cell types in the cerebellum…” I would argue that the sequencing has given us more information on the actual genes themselves, I am not sure that we have learned much more about patterning per se (at least not yet). This claim needs to be toned down.

We have toned down this statement to ‘has provided extensive information about the gene expression of distinct cell types in the cerebellum’ as requested by the reviewer.

4) In the statement above, the sentence should read “…information about the…”. The word “about” is missing.

We thank the reviewer for spotting this error. We have added the missing ‘about’.

5) The authors state “The cerebellum forms at the midbrain-hindbrain boundary…” I think this is not quite accurate. The cerebellum develops from the hindbrain, which is adjacent to the boundary.

We have rephrased this sentence to ‘The cerebellum forms from rhombomere 1 adjacent to the midbrain-hindbrain boundary’ to be more precise.

6) The cardinal lobes of the cerebellum should be labeled in Figure 1D.

We have re-drawn Figure 1D to more accurately reflect the P4 cerebellum and added labels for distinct lobules.

7) The authors state “…The anterior cerebellum frequently shows greater vulnerability and cell death,…” I think this issue requires a more in-depth discussion. It’s not quite so simple because many such models that describe anterior cerebellum defects do so based on sagittal sections. This of course ignores the zonal patterning effects. And because these descriptions are mainly from animal models of ataxia, which tend to have the same patterning issues, it seems like the anterior cerebellum is more affected. But in fact there is a bias in what has been described and how it has been described. The authors correctly allude to the nervous mutant, which has a more pronounced degeneration in zebrin positive cells, as opposed to the ataxia mutants which have degeneration in zebrin negative cells (making the anterior seem more affected, but in its really an issue of zonal patterning rather an issue of anterior versus posterior). Therefore, the authors are certainly not wrong in their statements, but this idea needs to be fully discussed so that those not familiar with the details get a full appreciation of the patterning and pathology.

In response to the reviewer’s suggestion, we have rephrased this statement and added additional sentences to the paragraph to emphasize that the observed vulnerability is likely to be a result of a particular molecular pattern rather than anterior vs posterior zone.

8) Given that zones/zebrin patterning is at the heart of this review, the authors would serve the reader well by providing a bigger and more complete schematic of the zebrin map. Figure 3A is very hard to interpret. No stripe specific labels are provided, and the labels for AZ, CZ, PZ, and NZ are hard to match for the hemispheres and vermis since an unfolded map is presented.

We have modified Figure 3A and moved zone labels to the left side of the figure for more clarity. We have chosen not to include specific stripe labels as this is not focused on in this review and has been described in detail elsewhere. References to a full Zebrin map have been added to the text and figure legend.

9) Figure 3B needs some labels.

In response to comments from all reviewers, we have removed Figure 3B.

10) For the middle panel of Figure 3D, why not just show both sides of the cerebellum otherwise the coordinates in the right panel are very confusing to interpret.

We have clarified the description of Figure 3 D (now C) to emphasize that only one hemisphere is depicted. We feel that showing both hemispheres would be rather repetitive and take up unnecessary space in the figure.

11) The authors state “the adult cerebellum is patterned by vertical parasagittal stripes”. I am not sure I have ever heard them described as vertical. It might be better to use previous language otherwise readers are likely to get confused.

We apologise for the confusion and have removed the word ‘vertical’ from this sentence.

12) I am not sure I understand what the authors mean by “Reelin-dependent ‘change of posture’.

We have rephrased this sentence to ‘Reelin-dependent change in direction to radial migration’ to provide better clarity.

13) There are several papers from Alexandra Joyner’s lab regarding Engrailed1 and Engrailed2 patterns and how mutations in these genes impact stripe patterning that should be cited, potentially in the second paragraph of page 12.

As suggested by the reviewer, we have included references to En1/2 patterning on page 12.

14) In Figure 4A, what is meant by a “variable” PC marker? I suspect variable isn’t the perfect choice of word.

We have relabelled Figure 4A to distinguish between broadly expressed PC markers including Foxp2, Skor2, and Rora, and variably expressed PC cluster markers that were defined in previous imaging studies.

15) The authors state “To date, many studies investigating the physiological properties of Purkinje cells focus only on the binary division of Z+ and Z- Purkinje cells.”. True, but I think it is also fair to mention the papers that don’t only focus in this. The authors should reference some of the work by Richard Hawkes (p-path, HNK1) and Sillitoe (NFH).

We have added references to these markers in an earlier paragraph (lines 329-342), where this information seems more relevant. To date, physiological studies have indeed focused on Z+ and Z- Purkinje cells, and the physiological properties of other Purkinje subtypes have not yet been characterised.

Reviewer 2 Report

In this manuscript, Apsley and Becker summarize and compare the recent single-cell sequencing data on the developing and adult cerebellum from various organisms from the Purkinje cell perspective. The review is well-written and would be of great interest to the field. I have two minor suggestions. 

1)    I’m having a hard time understanding the value of the data presented in Table 2. I understand that it is the significantly enriched genes in the human PCs compared to the rest of the dataset. However, I think a table summarising the common and unique genes between human and mouse PCs could be more instructive in the context of this paper. Table 2 and an equivalent for the mouse data set can then be added as a supplementary file. 

2)    Figure 3 can benefit from some legends. Particularly, it is hard to understand panels B and D. Are the clusters demonstrated in D hypothetical or do they present the data presented in E? If the latter is correct, it would be nice to label some of the known genes, even if the full spectrum of molecular subtypes is not known. 

Author Response

Reviewer #2

In this manuscript, Apsley and Becker summarize and compare the recent single-cell sequencing data on the developing and adult cerebellum from various organisms from the Purkinje cell perspective. The review is well-written and would be of great interest to the field. I have two minor suggestions. 

1) I’m having a hard time understanding the value of the data presented in Table 2. I understand that it is the significantly enriched genes in the human PCs compared to the rest of the dataset. However, I think a table summarising the common and unique genes between human and mouse PCs could be more instructive in the context of this paper. Table 2 and an equivalent for the mouse data set can then be added as a supplementary file.

In response to the reviewer’s comment, we have modified the table legend to clarify that the table lists genes that have been identified across both species. Individual lists for mouse and human datasets are available in the respective papers, i.e., Carter et al (Figure S3) and Aldinger et al (Table S9).

2) Figure 3 can benefit from some legends. Particularly, it is hard to understand panels B and D. Are the clusters demonstrated in D hypothetical or do they present the data presented in E? If the latter is correct, it would be nice to label some of the known genes, even if the full spectrum of molecular subtypes is not known.

We have modified the legend for Figure 3. Panel B has been removed and the description of panels D and E (now C, D) has been edited to provide more detail.

Reviewer 3 Report

The paper is well written, informative, and of potential interest. The research and analysis performed are convincing and well presented. Minor suggestions below:

-       Ln 30: unclear sentence: Unlike the cerebral cortex, the cytoarchitecture of the cerebellar cortex is remarkably 30 uniform…”

-       Fig2 and Table 1 overlap

-       Human genes should be in ITALIZED UPPERCASE letters, while the mouse genes – in Capitalized Italized letters. This will be helpful as large part of the paper compares human to mouse settings.

-       The legend of Fig3  is brief and unclear – i.e. what do the colors show in 3a, 3b, 3c and 3d?

-       Figure 4 is low resolution and not legit.

-       In section 3.3 “Understanding Purkinje cell identity using single-cell transcriptomics” it would be good to discuss emerging approaches for assessment of cell heterogeneity and plasticity such as identification of expressed SNP/SNVs (with either DNA- or RNA-origin) from scRNA-seq data (i.e. PMID: 31504520, PMID: 34680953).

Author Response

Reviewer #3

The paper is well written, informative, and of potential interest. The research and analysis performed are convincing and well presented. Minor suggestions below:

- Ln 30: unclear sentence: “Unlike the cerebral cortex, the cytoarchitecture of the cerebellar cortex is remarkably 30 uniform…”

We have checked this sentence, which appears to be correct in the current version of the Word document. We think that the reviewer might have experienced some formatting issues when opening the document on their computer. We will submit also a pdf of the manuscript to avoid formatting issues.

- Fig2 and Table 1 overlap

Please see comment above.

- Human genes should be in ITALIZED UPPERCASE letters, while the mouse genes – in Capitalized Italized letters. This will be helpful as large part of the paper compares human to mouse settings.

We have carefully checked the manuscript to ensure genes are appropriately annotated.

- The legend of Fig3  is brief and unclear – i.e. what do the colors show in 3a, 3b, 3c and 3d?

We have modified the legend for Figure 3 to more clearly describe the depicted images.

- Figure 4 is low resolution and not legit.

We believe this might be due to formatting issues that the reviewer experienced. However, we will provide high-resolution images for all figures in the revision process.

- In section 3.3 “Understanding Purkinje cell identity using single-cell transcriptomics” it would be good to discuss emerging approaches for assessment of cell heterogeneity and plasticity such as identification of expressed SNP/SNVs (with either DNA- or RNA-origin) from scRNA-seq data (i.e. PMID: 31504520, PMID: 34680953).

We thank the reviewer for highlighting these methods. We think that these would be most relevant to discuss in the context of disease but not for the developmental single-cell sequencing as is the focus of this review.